# Identification, Expression and Antimicrobial Functional Analysis of Interleukin-8 (IL-8) in Response to *Streptococcus iniae* and *Flavobacterium covae* in Asian Seabass (*Lates calcarifer* Bloch, 1790)

**DOI:** 10.3390/ani14030475

**Published:** 2024-01-31

**Authors:** Chayanee Muangrerk, Anurak Uchuwittayakul, Prapansak Srisapoome

**Affiliations:** 1Laboratory of Aquatic Animal Health Management, Department of Aquaculture, Faculty of Fisheries, Kasetsart University, 50 Paholayothin Road, Ladyao, Chatuchak, Bangkok 10900, Thailand; chayanee.mu@ku.th (C.M.); ffisarb@ku.ac.th (A.U.); 2Center of Excellence in Aquatic Animal Health Management, Department of Aquaculture, Faculty of Fisheries, Kasetsart University, 50 Paholayothin Road, Ladyao, Chatuchak, Bangkok 10900, Thailand

**Keywords:** proinflammatory cytokine, interleukin-8, Asian seabass, *Streptococcus iniae*, *Flavobacterium covae*, antimicrobial activity

## Abstract

**Simple Summary:**

As a proinflammatory cytokine, interleukin-8 (IL-8) plays a crucial function in inflammatory responses by recruiting and regulating monocytes and lymphocytes during the early stages of inflammation. In this study, a cDNA encoding the Asian seabass (*Lates calcarifer*) *IL-8* gene was cloned and referred to as *LcIL-8*. Moreover, the expression levels of *LcIL-8* in various tissues of normal and diseased fish were analyzed by means of qRT–PCR. Additionally, the recombinant *Lc*IL-8 protein (r*Lc*IL-8) was overexpressed in the *Escherichia coli* system, and its biological functions under various conditions were investigated. *LcIL-8* transcripts were expressed in all the tested tissues of normal Asian seabass. The constitutive mRNA expression of *LcIL-8* was quantified in six tissues of fish infected with virulent *Streptococcus iniae* and *Flavobacterium covae* at three different concentrations. The minimum inhibitory concentration (MIC) and therapeutic effects of the r*Lc*IL-8 protein against *S. iniae* were also thoroughly investigated. The obtained findings could be used to further develop prophylactic or therapeutic strategies applicable to the Asian seabass farming industry.

**Abstract:**

In this research, the proinflammatory cytokine interleukin-8 (IL-8) was shown to play a key role in inflammatory responses in fish. This study involved the cloning of the gene that encodes IL-8 in Asian seabass (*Lates calcarifer*) as well as analyses of its expression and function in this fish. The expression levels of *LcIL-8* indicated that it was broadly expressed in most analyzed tissues, with the most predominant expression in the whole blood 6 to 24 h after infection with *S. iniae* at concentrations of 10^5^ colony-forming units (CFU)/fish (*p* < 0.05). After fish were immersed in *F. covae*, the *LcIL-8* transcript was upregulated in the gills, liver and intestine, and the highest expression level was observed in the gills. However, *LcIL-8* was downregulated in all the tested tissues at 48 and 96 h after infection with the two pathogenic strains, indicating that *Lc*-*IL8* has a short half-life during the early immune responses to pathogens. Moreover, the MIC of the r*Lc*IL-8 protein against *S. iniae* was 10.42 ± 3.61 µg/mL. Furthermore, functional analyses clearly demonstrated that 10 and 100 µg of the r*Lc*IL-8 protein efficiently enhanced the phagocytic activity of Asian seabass phagocytes in vitro (*p* < 0.05). Additionally, in vivo injection of *S. iniae* following the r*Lc*IL-8 protein indicated that 50 and 100 µg of r*Lc*-IL-8 were highly effective in protecting fish from this pathogen (*p* < 0.001). The obtained results demonstrate that r*Lc*IL-8 possesses a biological function in the defense against bacterial infections in Asian seabass.

## 1. Introduction

Asian seabass or *Lates calcarifer* (Bloch, 1790) is considered an economically significant fish in Thailand and several other countries in the Asian region. The major export markets for Asian seabass include China, Germany, the Netherlands, Indonesia, and Vietnam [1]. Thailand is the 16th largest exporter of Asian seabass in the world. Currently, Asian seabass aquaculture has expanded throughout Thailand, becoming an intensive culture system. However, this system causes poor water quality and rapid changes in the aquatic environment during certain stages of cultivation. These effects directly increase stress in seabass and induce susceptibility to severe diseases caused by various parasites, viruses, and bacteria. Recently, in Thailand, major pathogenic bacteria, including *Aeromonas* spp., *Flavobacterium* spp., *Vibrio* spp., and *Streptococcus* spp. have been identified in Asian seabass farms [2,3,4,5]. Specifically, *S. iniae* and *F. covae*, which are the causative agents of streptococcosis and columnaris, respectively, have been identified in Asian seabass farms. Clinical signs of these diseases, such as skin hemorrhage, fin rot, and lethargy, are rapidly developed and can lead to fish mortality within 24–72 h, resulting in substantial losses for the Asian seabass farming industry in Thailand [6,7]. The utilization of antibiotic and chemical treatments for disease management in these fish has been on the decline, a trend primarily attributed to concerns about the potential health risks for consumers, compounded by the association of many fish pathogens with the development of antibiotic resistance [8].

To address these problems and sustain Asian seabass aquaculture, an evaluation of the molecular characterization and expression responses of immune-related genes is needed for the advanced development of therapeutic and prophylactic strategies. Based on current knowledge, it is known that the immunity of fish, especially teleost fish, can be divided into innate and specific or acquired immunity. The innate immunity is the first-line immune defense against harmful pathogens that attempt to enter the bodies of all animals [9]. It has a nonspecific manner, has limited recognition, and responds rapidly to several microorganisms. This system can be divided into three crucial components: physical, cellular and humoral defenses. Both humoral and cellular defenses are crucial components that are typically closely associated with the specific immune system. In particular, the humoral response depends upon the activity of several molecules, such as cytokines, complements, transferrin, and lytic enzymes [10]. Cytokines are a group of molecules that certain cells produce to perform important functions in cell signaling during wound-healing, inflammation and pathogen infections, especially those caused by bacteria or viruses [11]. In the inflammatory response process, cytokines that promote the initial stages of inflammation are also known as “proinflammatory cytokines”. These molecules are not secreted under ordinary conditions. However, they are secreted when the immune system is triggered by pathogens and classified as a small cytokine family named “chemokine”, which performs a variety of functions, including chemotaxis, T-cell function and differentiation, as well as angiogenesis [12]. Among these molecules, interleukin-8 (IL-8), referred to as “CXCL8 or neutrophil-activating peptide (NAP-1)”, was the first chemokine to be identified and was initially identified from human blood monocytes stimulated with lipopolysaccharide (LPS) [13]. Currently, it is widely acknowledged that IL-8 is produced by various immune and nonimmune cells, such as fibroblasts and endothelial cells, in response to a wide range of pathogenic stimuli [14]. It is a pivotal chemokine involved in the proinflammatory process. This chemokine belongs to the subfamily of CXC chemokines and serves as a chemotactic factor, attracting neutrophils and macrophages during the early stages of inflammation. Moreover, it possesses the capability to stimulate cells by initiating respiratory bursts, degranulation of storage proteins and production of lipid mediators [15,16,17]. The ability of IL-8 to attract specific cells is ascribed to the absence of a Glu-Leu-Arg (ELR) motif next to a CXC sequence at the N-terminal part [18]. However, in certain fish species, such as Atlantic cod and haddock, the presence of the ELR motif has been discovered to have the capacity to attract PMN cells, especially during inflammation. In fish, the first IL-8 was identified in lamprey (*Lampetra fluviatilis*) [19]. Subsequently, this molecule has been characterized in several other fish species, such as Japanese flounder [20], rainbow trout [21,22], common carp [23], catfish [24], zebrafish [25] and Atlantic cod [26].

Our understanding of the immune function and regulatory processes governing IL-8 in Asian seabass is currently limited. Therefore, this study aims to investigate the structure of the complementary DNA (cDNA) of the *IL-8* gene and its transcriptional-level expression under pathogenic bacterial infection. Additionally, it aims to elucidate the function of this molecule under various conditions. The insights obtained from the current research are important for elucidating the response mechanisms and biological functions of *Lc*IL-8 to inform the development of therapeutic and prophylactic methods that elevate disease resistance against pathogenic bacteria in the Asian seabass aquaculture industry.

## 2. Materials and Methods

### 2.1. Experimental Animals

Healthy Asian seabasses that were free from specific pathogens such as viruses and bacteria and weighed 10–12 g were purchased from a local farm in Chachoengsao Province, Thailand. The fish were weekly acclimatized at the Department of Aquaculture, Faculty of Fisheries, Kasetsart University (Bangkok, Thailand). The experimental fish were further raised in a quarantined 3000-L fiberglass tank containing 2000 L of clean dechlorinated water that was adjusted to a salinity of 10 ppt with a full aeration system. The fish were maintained at a temperature of 28–30 °C, pH 7–8, a dissolved oxygen content (DO) of 5–7 mg/L, and a pH ranging from 7.5 to 8.5. Additionally, the alkalinity was maintained within 100–150 mg/L as the CaCO_3_ range. During the acclimatization period, the experimental fish were fed twice a day with Seabass feed C-5003 (Uni-President, Di An, Vietnam) commercial pellet feed (the nutrient composition consisted of 44% protein, 7% crude fat, 3% crude fiber, and 11% moisture) at 5% body weight for 1 week before the beginning of the experiment.

### 2.2. Cloning and Characterization of the cDNA Encoding the Mature LcIL-8 Protein of Asian Seabass

A healthy Nile tilapia from Section 2.1 was randomly selected. The total RNA was extracted from an approximately 25 mg sample of the head kidney of the euthanized fish using TRIzol reagent (Invitrogen, Waltham, MA, USA) following the manufacturer’s instructions. First-strand cDNA was subsequently synthesized from the isolated total RNA using ReverTra Ace^TM^ qPCR RT Master Mix with gDNA Remover (TOYOBO Bio-Technology, CO., LTD., Osaka, Japan) to eliminate contaminating genomic DNA. The nucleotide sequence of the open reading frame (ORF) of Asian seabass IL-8 was retrieved from the GenBank database (XM_018695863.2) (http://www.ncbi.nlm.nih.gov accessed on 9 August 2022) and cloned via PCR amplification. Specific primers were designed to amplify the sequence encoding *Lc*-IL8 starting from the beginning of the leader sequence to the stop codon (*Lc*IL-8 F and *Lc*IL-8 R, Table 1). The cycling conditions were as follows: 95 °C for 5 min; 40 cycles of 95 °C for 30 s, 55 °C for 30 s, and 72 °C for 1 min; and an elongation step of 72 °C for 10 min. The obtained PCR products were separated via electrophoresis on a 1.5% agarose gel. The target PCR band was cut and further purified using a QIAquick^®^ gel extraction kit (Qiagen, Venlo, The Netherlands). The cloning and sequencing protocols were strictly followed based on a previous study [18]. Sequence and structural analyses of the cDNA and the amino acid sequence of *Lc*-IL8 were conducted using the BLASTN and BLASTX programs available through the National Center for Biotechnology Information (NCBI; http://www.ncbi.nlm.nih.gov accessed on 9 August 2022). The ExPASy Molecular Biology server (http://us.expasy.org accessed on 9 August 2022) was employed to identify the signal sequence of the target protein.

### 2.3. Homology and Phylogenetic Analysis of LcIL-8 and Various Interleukin-8 Genes of Other Vertebrates

MatGAT version 2.01 (http://www.angelfire.com/nj2/arabidopsis/MatGAT.html, accessed on 14 October 2022) was used to assess the amino acid sequence identity and similarity of *Lc*IL-8 cDNA and other reported IL-8 gene sequences. To investigate the evolutionary relationships, a phylogenetic tree of *IL-8* genes from various vertebrate species was constructed. This tree was generated by aligning the amino acid sequences of *Lc*IL-8 cDNA with those of known *IL-8* genes from diverse vertebrate species using ClustalW (http://ebi.ac.uk/Tools/clustalw/index.html, accessed on 14 October 2022). The evolutionary tree of IL-8 molecules of vertebrates was constructed using the neighbor-joining method in MEGA 11.0 software (www.megasoftware.net, accessed on 14 October 2022) with 1000 bootstrap analyses.

### 2.4. Expression Analysis of LcIL-8 Gene in Various Tissues of Healthy Fish Using Quantitative Reverse-Transcription Real-Time PCR (qRT–PCR)

#### 2.4.1. Isolation of Total RNA and 1st Strand cDNA Synthesis

Four fish that were randomly selected in the previous section were used for the tissue extraction. Approximately 25 mg samples were harvested from the following 12 organs: brain (BR), gills (GL), heart (HR), liver (LV), spleen (SP), head kidney (HK), trunk kidney (TK), intestine (INT), stomach (SM), muscle (MC), skin (SK), and whole blood (WB). Whole blood was obtained via withdrawal from the caudal vein at caudal peduncle areas using a 1 mL syringe equipped with a heparinized 23-G needle. The total RNA was extracted from these 12 tissues (25 mg) using TRIzol reagent (Invitrogen, Waltham, MA, USA) according to the manufacturer’s protocol. The precipitated RNA pellets from the target tissues were dried in air, and sterile nuclease-free water was used to dissolve the isolated total RNA. The quantity and quality of the obtained RNA were identified based on an absorbance ratio of OD260/280 nm using NanoDrop™ 2000/2000c Spectrophotometers (Thermo Fisher Scientific, Waltham, MA, USA). Subsequently, 1 µg of total RNA from each tissue was separately used to synthesize the 1st strand cDNA using ReverTra Ace^TM^ qPCR RT Master Mix with gDNA Remover (TOYOBO Bio-Technology, CO., LTD., Osaka, Japan). The 1st strand cDNA from all the tissues was synthesized based on the protocol recommended by the company and immediately stored at −20 °C in a deep freezer.

#### 2.4.2. qRT–PCR Analysis

To assess the expression levels of the *LcIL-8* transcripts in the 12 tissues of healthy fish from the above section, qRT–PCR techniques were conducted with the specific primers *Lc*IL-8 qF and *Lc*IL-8 qR (Table 1). One microgram of first-strand cDNA from every organ was employed using Brilliant III Ultra-Fast SYBR^®^ Green qRT–PCR Master Mix (Agilent Technologies, Inc., Santa Clara, CA, USA), which was performed using an Mx3005P real-time PCR system (Stratagene, San Diego, CA, USA). The analytical software version 4.0 was employed to reveal the gene expression levels, following the manufacturer’s recommended protocol. Each PCR reaction was performed in a total volume of 25 µL consisting 1.0 µL of 1st strand cDNA, 12.5 µL of Brilliant III Ultra-Fast SYBR Green qRT–PCR master mix, 9.5 µL of sterile distilled water and 1.0 µL of each *Lc*IL-8 qF and *Lc*IL-8 qR primer (Table 1), resulting in final concentrations of 10 µM. The efficiency of qPCR for each primer set was assessed by means of comparison with a standard curve prior to the experiment, with a target range of 95–110%. The PCR cycling reaction was set as follows: an initial step of denaturation at 95 °C for 10 min, followed by 40 cycles of denaturation at the same temperature for a precisely timed 30 s, annealing at 56 °C for 30 s, and extension at 72 °C for 1 min. To standardize the obtained results and eliminate variations in the mRNA and cDNA, the transcriptional level of the *β-actin* gene was used as the internal control, with the primers *Lc*-β-actin qF and *Lc*-β-actin qF (Table 1). The primer specificity was confirmed through DNA melting curve analysis. Triplicate PCR reactions were conducted for both the *β-actin* and *LcIL-8* genes in each sample. All the primers for the gene expression analysis were designed based on the sequences indicated in the Asian seabass genome project (ASM164080v1). A reference plasmid of *β-actin* and *LcIL-8* genes was 10-fold serially diluted to conduct calibration curves to analyze their PCR efficiency. The relative copy number of the target mRNA was calculated using 2^−ΔΔCT^ analysis, and the expression ratio in the muscle was used as a calibrator [27].

### 2.5. Response Analysis of LcIL-8 under Stimulation with S. iniae and F. covae using qRT–PCR

#### 2.5.1. Bacterial Strains and Preparation

The pathogenic bacterial strains *S. iniae* and *F. covae* were obtained from the LAAHM. A single colony of *S. iniae* was identified as described by [28]. The bacteria were cultured in 100 mL of trypticase soy broth (TSB) medium and further incubated in a shaking incubator at 30 °C for 24 h. Subsequently, the bacteria were harvested via centrifugation at 2300× *g* for 15 min and then resuspended in 0.85% NaCl solution until the absorbance at an optical density (OD) of 600 reached 1.00, resulting in a concentration of 1 × 10^9^ colony-forming units (CFU)/mL. Then, the same solution was diluted to obtain final concentrations of 1 × 10^2^, 1 × 10^4^ and 1 × 10^6^ CFU/mL. Similarly, a pure strain of *F. covae* was cultured in 200 mL of Shieh’s medium broth in an incubator shaker at 25–30 °C for 24 h. Then, the obtained bacterial culture was inoculated into 1 L of Shieh’s broth and incubated under the same conditions. Finally, the bacterial pellet was carefully resuspended in a 0.85% NaCl solution. The concentrations of *F. covae* were finally adjusted to 1 × 10^2^, 1 × 10^4^ and 1 × 10^6^ CFU/mL for the other experiments below.

#### 2.5.2. Animals and Experimental Design

One hundred and eighty Asian seabass from the previous section were selected and divided into six fiberglass tanks, each with a 250 L capacity (30 fish/tank). Tanks 1–3 were filled with salt water at a salinity of 5 ppt, while tanks 4–6 were filled with fresh water and further set up as prepared above for 7 days. After the acclimatization period, every healthy Asian seabass in tanks 1–3 was injected intraperitoneally (IP) with 0.1 mL of *S. iniae* solution prepared in 0.85% NaCl solution at 3 different concentrations of 1 × 10^2^, 1 × 10^4^ and 1 × 10^6^ CFU/mL, respectively. Meanwhile, for the fish in tanks 4–6, all the individuals were removed and placed into 10 L tanks containing clean freshwater, which was prepared with the *F. covae* bacterial solution and had the same concentration as the *S. iniae* solution. Immersion was carried out for a duration of 30 min, and then the fish were returned to each tank as before. Sampling was conducted at 0 (Control), 6, 12, 24, 48 and 96 h after IP injection; the head kidney, liver, spleen, gills and intestine were separately collected from 4 fish in each *F. covae* and *S. iniae* injected group.

#### 2.5.3. Preparation of Total RNA Extraction and 1st Strand cDNA

After infection, tissue samples were collected for the total RNA extraction with the methods described above. The total RNA templates were prepared at a concentration of 1000 ng/mL from each group at different time points. First-strand cDNA was subsequently synthesized using the same method described previously and stored at −20 °C until use.

#### 2.5.4. qRT–PCR Assay

First-strand cDNAs synthesized from the head kidney, liver, spleen, gills and intestine of the bacterial infected fish at each time point were analyzed using qRT–PCR as described above, albeit with slight modifications, where the relative expression levels of each β-*actin* and *LcIL-8* gene at h 0 were used as calibrators.

### 2.6. Overexpression, Production and Purification of Recombinant LcIL-8 Protein (rLcIL-8)

#### 2.6.1. Construction of Recombinant LcIL-8 DNA

The plasmid DNA containing the complete nucleotide sequence obtained in the experiment detailed in Section 2.2 was subjected to *Nde* I and *Xho* I double-restriction digestion according to the recommendations of Thermo Scientific’s protocol (Waltham, MA, USA). The resulting DNA bands were separated via electrophoresis on a 1.5% agarose gel. Subsequently, the target DNA fragments were purified using a QIAquick^®^ gel extraction kit (Qiagen, Venlo, Netherlands). These DNA fragments were linked with an *Xho* I/*Nde* I-cut pET28b (+) expression vector with an N-terminal 6× His-tag. The ligated recombinant plasmid was further transformed into prepared JM109 competent cells. The positively transformed bacterium was selected and cultured at 37 °C in Luria–Bertani (LB) broth supplemented with 100 µg/mL kanamycin. The transformed clone was isolated, preserved, and subjected to plasmid DNA extraction, following the previously described procedure. Subsequently, the r*Lc*IL-8 plasmid was retransformed into *E. coli* strain BL21 (DE3) using heat–cold shock techniques. The positive cells containing the r*Lc*IL-8 plasmid were sequenced using a similar method as previously mentioned to confirm the correctness of the *Lc*IL-8 sequence.

#### 2.6.2. Overexpression of Recombinant LcIL-8 Protein

A single positive colony of BL21 cells containing r*Lc*IL-8, as confirmed by means of nucleotide sequencing, was inoculated in 10 mL of LB broth supplemented with kanamycin. The culture was allowed to incubate overnight in a water bath shaker at 37 °C. Then, 1 mL of prepared preculture bacterial solution was transferred into 10 mL of LB broth supplementing kanamycin and incubated in a shaking incubator until the absorbance of the bacterial solution was 0.6 at OD 600 nm. To induce overexpression, 1.0 mM IPTG was included to the bacterial solution. The bacterial solution was then cultivated continuously for a period of 24 h. During the incubation period, samples were collected at specific time points: 0, 4, 12, and 24 h post-IPTG induction, and each 1 mL aliquot of the bacterial culture was subjected to centrifugation at 7300× *g* for 5 min to facilitate the optimization of the protein expression conditions. The expression of recombinant proteins at different time intervals was identified using sodium dodecyl sulfate–polyacrylamide gel electrophoresis (SDS–PAGE) (Thermo Fisher Scientific, Waltham, MA, USA).

#### 2.6.3. Purification of Recombinant LcIL-8

The induced recombinant protein (r*Lc*IL-8) was extracted using a HiTrap™ Protein A HP column (GE Healthcare, Chicago, NJ, USA) and an ÄKTA pure protein purification kit (ÄKTA pure 25 L, GE Healthcare, NJ, USA) following the company’s protocols, with a modified protocol designed for inclusion body proteins. Briefly, a 100 mL aliquot of the bacterial solution was centrifuged at 2300× *g* for 5 min to harvest bacterial cells. The bacterial cells were then resuspended in 20 mM Tris-HCl (pH 8.0) with a total volume of 10 mL and subjected to an iced sonication system. Following sonication, the suspension was vigorously centrifuged at 12,000× *g* for approximately 10 min to collect the bacterial inclusion bodies. These components were subsequently dissolved using 15 mL of solubilization buffer composed of 20 mM Tris-HCl, 0.5 M NaCl, 5 mM imidazole, 6 M urea, and 1 mM 2-mercaptoethanol (pH 8.0). After purification, the recombinant protein was eluted with a linear gradient of 20–500 mM imidazole within an elution buffer (pH 8.0). The resulting protein fractions were collected within fresh Eppendorf tubes and subjected to analysis using SDS–PAGE techniques, as previously described. The obtained recombinant protein was dialyzed against a solution containing 150 mM NaCl, 20 mM Tris-HCl, pH 8.0 and 20% glycerol, with this process occurring overnight at 4 °C to eliminate small unwanted molecules, such as 2-mercaptoethanol, imidazole and urea. To determine the protein concentrations, the Bradford protein assay was employed, and the obtained results were compared to standard serial twofold protein dilutions starting from 2 mg/mL albumin. The quantification of all the concentrations was conducted by measuring the absorbance at OD595 nm utilizing an iMark™ Microplate Absorbance Reader (Bio-Rad, Hercules, CA, USA).

#### 2.6.4. Western Blot Analysis

Western blot assays were carried out to separate and determine the molecular weight of the target r*Lc*IL-8 protein. The r*Lc*IL-8 protein was run on a 12% SDS–PAGE gel and subsequently electrophoretically transferred onto a PVDF (polyvinylidene difluoride) membrane using the Invitrogen™ Power Blotter XL System (Invitrogen, Waltham, MA, USA) with transfer buffer at 100 V for 15 min. Following the transfer step, the protein-containing membrane was blocked with a one-step blocking solution and incubated with a 1:5000 dilution of an anti-6× His tag^®^ antibody solution (Thermo Fisher Scientific, Waltham, MA, USA) for at least 2 h at room temperature (RT). The membrane was further subjected to three washes with 0.05% Tween-TST. The hybridized protein was visualized using UltraScence Pico Ultra Western Substrate (Bio-Helix, Xinbei, Taiwan) for 1–2 min until color appeared, enabling the identification of specific bands corresponding to the r*Lc*IL-8 protein on the PVDF membrane.

### 2.7. Effects of rLcIL-8 Protein on Phagocytic Activity (In Vitro)

A phagocytic activity was chosen to investigate the immune function of the r*Lc*IL-8 protein, which was modified by Bunnoy et al. (2023) [28]. In this experiment, white blood cells were isolated from approximately 100 g of 4 healthy Asian seabasses using a similar method to that described above. A volume of 1 mL of whole blood was transferred into a 15 mL PE tube and 2 mL of RPMI 1640 medium (Caissonlabs, North Logan, UT, USA) was carefully added. After gentle mixing, the mixture was then loaded into a new PE tube, which contained 3 mL of Lymphoprep (Dayang Chem Co., Ltd., Hangzhou, China). The subsequent step involved centrifugation in a swing rotor Benchmark Scientific Z216-CMB Z216 (Swedesboro, NJ, USA) at 600× *g* at 25 °C for 30 min. The A band, which consisted of lymphocytes and monocytes, was pipetted in 1 mL, mixed with 2 mL of Roswell Park Memorial Institute (RPMI) medium, and subjected to centrifugation at 300× *g* at 25 °C for 15 min, which was repeated three times. Following dilution, the white blood cell pellets were collected to a final density of 5 × 10^6^ cells/mL. Subsequently, 200 µL of the prepared white blood cell suspension was carefully loaded onto 12 cover slips, and these phagocytes were settled to attach to the surface of cover glasses over a 2 h incubation period. Meanwhile, green fluorescence microlatex beads (2 μm) (Sigma Aldrich, St Louis, MO, USA) were diluted to a concentration of 1 × 10^8^ beads/mL using RPMI medium. Four experimental groups were created for each r*Lc*IL-8 protein concentration. In groups 1, 2, and 3, the beads were carefully coated with 1, 10 and 100 µg/mL in RPMI medium, respectively. The uncoated beads in group 4 (control) were solely immersed with RPMI 1640 medium. This experiment was conducted in triplicate for each treatment.

After 2 h of incubation, unattached phagocytes were gradually washed in three rounds with RPMI medium. Subsequently, 200 µL of the four latex bead groups, each incubated with different r*Lc*IL-8 protein concentrations, were added to the cover glasses. The phagocytes and latex beads in each group were co-incubated at RT for 1.5 h to initiate the phagocytosis process. After 1.5 h, the excess beads and unattached cells were carefully cleaned up three times with RPMI medium. All the remaining cells on the cover glass were imbued with methylene blue and eosin (Dip-Quick Staining dye) (Thermo Fisher Scientific, Waltham, MA, USA). Subsequently, a minimum of 200 phagocytic cells were observed using a brightfield microscope. The phagocytic index (PI) and the percentage of phagocytosis (PA) were calculated according to the methods described by [28].

### 2.8. Minimum Inhibitory Concentration (MIC) of rLcIL-8 Protein (In Vitro)

In this part, *S. iniae* was only selected based on its systemic infection properties, and for the challenge experiment, it was prepared as detailed in Section 2.5.1 and was separately employed to determine the minimal inhibitory concentrations (MICs) using micro-broth dilution methods. The r*Lc*IL-8 protein was serially diluted twofold from 100 to 0.195 µg/mL (in triplicate) using Mueller–Hinton (MH) broth in a U-shape bottom 96-well microtiter plate. Subsequently, 100 µL of the *S. iniae* solution at a concentration of 1 × 10^5^ CFU/mL was added (1 × 10^4^ CFU per well), and the total volume in each well was adjusted to 200 µL. Wells containing only bacteria in MH broth were designated as a negative control, while the positive control group contained only MH broth. Following a 24 h incubation at 30 °C, the lowest concentration that exhibited clear broth was defined as the MIC. Subsequently, the solutions in every observed well were pipetted and spread onto MH agar. The plates were incubated under the same conditions, and bacterial growth was monitored via colony counting within the range of 30–300 colonies. The bacterial quantity was calculated as colony-forming units per milliliter (CFU/mL) to assess the inhibitory effect on bacterial growth.

### 2.9. Effects of the rLcIL-8 Protein on Resistance to S. iniae in Asian Seabass

#### 2.9.1. Experimental Conditions

The one hundred and twenty healthy Asian seabass (10 g) detailed in the previous section were used. The fish were randomly selected and placed into twelve separate 250-L fiberglass tanks, each containing 200 L of 5 ppt saline water (10 fish/tank). These tanks maintained the same environmental conditions as previously described. Four groups (3 tanks/group) were set for the further experiments below.

#### 2.9.2. Bacterial Challenge

A pathogenic *S. iniae* strain was prepared in 0.85% NaCl, as described above, to obtain a final concentration of 1 × 10^6^ CFU/mL. The experiment was conducted in triplicate. In each of the four groups, ten fish received intraperitoneal injections with 0.1 mL of a bacterial solution. After inoculation for 1 h, three r*Lc*IL-8 concentrations, including 10, 50 and 100 µg/mL, were injected under the same conditions. Every fish in each tank of each group was injected with 0.1 mL 0.85% NaCl to set as controls. Subsequently, the experimental fish in each group were reared for 14 days. During this period, the daily mortality was recorded, and any abnormal fish in each group were examined for *S. iniae* infection in the spleen and liver using the streak plate method on trypticase soy agar (TSA) to confirm *S. iniae* infection.

### 2.10. Statistical and Data Analysis

The relative expression ratio of the *LcIL-8* mRNAs was assessed at different time points via two-way analysis of variance (ANOVA) based on a 3 × 6 factorial design in a completely randomized design (CRD) of 2 factors of 3 bacterial concentrations and 6 detection periods. The percent phagocytosis (PA) and the phagocytic index (PI) were analyzed with a one-way ANOVA based on the CRD design. The significant difference in the means of the target parameters was further determined using Duncan’s new multiple range test (DMRT) method with SPSS software version 20.0 (IBM SPSS Statistics 24.0). The data presupposition of the homogeneity and normal distributions of all the obtained variance were cautiously determined before statistic investigation. The interactions of the 2 target factors were analyzed throughout. A significance level of *p* < 0.05 was statistically considered.

A survival analysis of the Asian seabass in the *S. iniae* challenge tests in each group was investigated using the Kaplan–Meier method. The significance levels between the control and treatments were indicated as * *p* < 0.05 and *** *p* < 0.001 using Student’s *t*-test.

## 3. Results

### 3.1. Characterization of the LcIL-8 cDNA

After obtaining the mature mRNA of the *LcIL-8* gene from cloning, a comparison with the GenBank database revealed its 100% identity to be the full-length cDNA encoding the interleukin-8 gene of Asian seabass (GenBank accession No. XM_018695863.2). This sequence was named *Lc*IL-8. The *Lc*IL-8 nucleotide sequences contained 237 bp, equal to 79 amino acids (Figure 1A,B). Mature *Lc*IL-8 exhibited a signature arrangement of four cysteines (C30, C32, C56, and C73), two of which were separated by an arginine residue, as observed in the CXC chemokine signature of other vertebrate CXC chemokines. However, the Glu-Leu-Arg (ELR) structure upstream of the CXC motif was not found in *Lc*IL-8 and was replaced with the ILR (Ile-Leu-Arg) motif. A protein structure analysis of the *Lc*IL-8 showed that it had a molecular weight and a theoretical isoelectric point (p*I*) of 12.2 kDa and 8.44, respectively.

### 3.2. Homology and Phylogenetic Analysis of IL-8 Genes in Vertebrates

Homology analyses of the observed amino acid sequences demonstrated that *Lc*IL-8 exhibited 24.8–76.6% sequence identity with *IL-8* genes in various higher vertebrates and fish species. The amino acid sequence identity and similarity of *Lc*IL-8 in Asian seabass and other organisms showed that the highest levels of both identity and similarity were shared with yellowfin seabream (*Acanthopagrus latus*), exhibiting scores of 76.6% and 86.2%, respectively (Appendix A).

The phylogenetic tree of the Asian seabass IL-8 protein (*Lc*IL-8), along with other IL-8 proteins of vertebrates, was constructed using reported information from the GenBank database. The tree included 37 IL-8 proteins from various vertebrate species obtained from 23 fish species and 14 higher vertebrates. The results revealed the three evolutionary clades of IL-8 proteins. Clade 1 contained five different subgroups of mammals, birds, Western clawed frogs, cartilaginous fish and European seabass IL-8 proteins, while clades 2 and 3 were associated with the teleost fish IL-8 group. *Lc*IL-8 was subclustered within the fish IL-8 first subgroup of clade 2 and exhibited close evolutionary relationships with yellowfin seabream (*Acanthopagrus latus*) and yellow croaker (*Larimichthys crocea*) (Figure 2).

### 3.3. qRT–PCR Analysis of LcIL-8 Transcripts in Various Tissues of Healthy Asian Seabass

The distribution of the *LcIL-8* transcript in healthy Asian seabass tissues was assessed using qRT–PCR techniques in 12 different tissues from 4 normal fish. The expression analysis revealed constitutive expression of *LcIL-8* among all the tested tissues. Remarkably, the highest expression level was found in the gills, at a level 18.28 ± 0.63-fold greater than in the muscle (*p* < 0.05). Additionally, moderate expression levels were observed in the intestine, spleen and stomach at 10.46 ± 0.31, 9.73 ± 0.14 and 6.06 ± 0.53-fold, respectively). In contrast, *LcIL-8* mRNA exhibited relatively low expression levels in the muscle, whole blood, heart, liver, head kidney, trunk kidney, skin, and brain (Figure 3).

### 3.4. Analysis of LcIL-8 Expression in Response to Different Concentrations of S. iniae and F. covae Using qRT–PCR

#### 3.4.1. Expression Level Analysis after *S. iniae* Infection

The qRT–PCR assay technique was used to quantify the *LcIL-8* transcriptional levels in Asian seabass after injection with three different concentrations of *S. iniae* at various time intervals. In whole blood (Figure 4A), a significant upregulation of the *LcIL-8* mRNA levels was observed in fish injected with 1 × 10^5^ CFU/fish *S. iniae* from 6 to 24 h, resulting in relative expression levels of 5.44 ± 0.21, 6.20 ± 0.18, and 9.27 ± 0.09, respectively (*p* < 0.05). This corresponds to an overall trend in the expression levels, indicating a significant upregulation of *LcIL-8* from 6 to 24 h for concentrations of 1 × 10^5^ CFU/fish. However, at 48 and 96 h, all the bacterial concentrations resulted in a significant decrease in the mRNA levels of *LcIL-8*. In the head kidney (Figure 4B), the concentrations of 1 × 10^5^ CFU/fish *S. iniae* significantly upregulated the *LcIL-8* mRNA levels at 12 h postinjection, with expression levels of 3.93 ± 0.37-fold (*p* < 0.05). Interestingly, at 48 and 96 h postinjection, *LcIL-8* was downregulated across all the bacterial concentrations, with extremely low expression levels.

In the liver (Figure 4C), a similar trend to that observed in the spleen (Figure 4D) at 24 h was noted, with the fish exposed to the highest concentration of bacteria exhibiting a 4.46 ± 0.41-fold upregulation of the target gene, while in the spleen, it was 3.14 ± 0.18-fold (*p* < 0.05). However, at 48 and 96 h, all the bacterial concentrations resulted in significantly lower expression levels compared to these peak levels (Figure 4C,D). Similarly, in the gills, the highest concentration of *S. iniae* induced a 3.43 ± 0.05-fold upregulation of *LcIL-8* expression at 6 h postinjection (*p* < 0.05) (Figure 4E). In the intestine, the *LcIL-8* transcript levels were significantly enhanced by a concentration of 1 × 10^3^ CFU/fish *S. iniae* at 12 and 24 h, which resulted in 2.63 ± 0.12 and 3.01 ± 0.37-fold changes, respectively (*p* < 0.05). An analysis of the overall time points and concentrations in the intestine revealed that the highest expression levels of *LcIL-8* were observed in fish injected with the moderate dose at 12 and 24 h (Figure 4F).

#### 3.4.2. Expression Level of LcIL-8 mRNA in Response to *F. covae* Infection

The qRT–PCR analysis of the *LcIL-8* mRNA levels in the six Asian seabass tissues in response to *F. covae* at different time points revealed that at 6 and 12 h, the highest bacterial concentration significantly induced higher mRNA expression of *LcIL-8* in all the tested tissues (*p* < 0.05). However, at 24, 48 and 96 h postinfection, this bacterial concentration resulted in significantly lower expression levels (Figure 5A–F). The mRNA expression was at the highest levels in the liver, gills and intestine (Figure 5C,E,F, respectively). In particular, high levels were observed in the gills at 12 h in fish exposed to 1 × 10^6^ CFU/mL *F. covae*, with levels of 11.14 ± 0.05-fold (*p* < 0.05). Interestingly, by 24 h, the expression levels of *LcIL-8* were downregulated and continued to decrease until 48 h postinfection for all the concentrations of *F. covae*. Subsequently, at 96 h, this gene was significantly upregulated again (Figure 5E). In the liver, the *LcIL-8* mRNA levels gradually increased from 6 to 12 h postinfection at both concentrations of bacteria (1 × 10^4^ and 1 × 10^6^ CFU/mL) and were downregulated to normal levels at 24 to 96 h postinfection (Figure 5C). Similarly, in the intestine, the mRNA expression level of the *LcIL-8* gene was obviously elevated at 6 h in response to concentrations of 1 × 10^6^ CFU/mL, with relative expression levels of 8.18 ± 0.12-fold (*p* < 0.05). However, at 12 h postinfection, *LcIL-8* gene expression gradually decreased and reached basal levels from 48 to 96 h postinfection at all the bacterial concentrations (Figure 5F).

In the whole blood (Figure 5A), similar to the trend observed in the spleen (Figure 5D), *LcIL-8* mRNA expression exhibited a clear and significant increase at 6 and 12 h postinfection. However, when fish were exposed to a bacterial concentration of 1 × 10^4^ CFU/mL, the *LcIL-8* mRNA levels were downregulated in the spleen, with a fold change of 0.97 ± 0.03, while elevated expression persisted exclusively in whole blood at 2.91 ± 0.25-fold (*p* < 0.05) (Figure 5A,D). In the head kidney, *LcIL-8* expression was clearly elevated from 6 to 12 h postinfection, with the maximal expression responding at 6 h in response to the highest bacterial concentrations, with fold changes of 4.69 ± 0.12 (*p* < 0.05). This trend was consistent with the overall time points and concentrations, with *LcIL-8* expression peaking at 6 h in fish injected with the highest dose (Figure 5B).

### 3.5. Overexpression of the Recombinant LcIL-8 Protein

The r*Lc*IL-8 was overexpressed in a bacterial expression system using *E. coli* BL21 cells. SDS–PAGE analysis revealed that the r*Lc*IL-8 protein was well expressed beginning at 12 h after IPTG induction, and it had an apparent molecular weight of approximately 12.2 kDa, which was consistent with the predicted molecular weight (Figure 6A). Notably, the protein was mostly found in inclusion bodies (IBs). After the IBs were successfully refolded using 6 M urea, the soluble form of the target protein was purified using the ÄKTA pure protein purification system. Further examination with Western blotting analysis confirmed the presence of the r*Lc*IL-8 protein, which was specifically recognized by an anti-6x His tag^®^ antibody (Figure 6B).

### 3.6. Effects of rLcIL-8 Protein on Phagocytic Activity and Phagocytic Index

The effects of r*Lc*IL-8 on the phagocytic activity (PA) and phagocytic index (PI) in Asian seabass phagocytes at three different concentrations are illustrated in Figure 7A–C. The results demonstrate that compared to the control group, the r*Lc*IL-8 group exhibited significantly enhanced PA at all the concentrations (1, 10 and 100 µg), with PA values of 46.33 ± 3.05, 47.33 ± 1.53, and 58.33 ± 4.16%, respectively (*p* < 0.05). Notably, the protein concentration of 100 µg exhibited strong PA-inducing ability (Figure 7A). However, the PI showed a significant increase only at protein concentrations of 10 and 100 µg in this group compared to the control, which resulted in indices of 1.56 ± 0.06 and 1.71 ± 0.16, respectively (*p* < 0.05) (Figure 7B).

### 3.7. MIC Investigations of the rLcIL-8 Protein against S. iniae

The efficacy of the r*Lc*IL-8 protein against the pathogenic *S. iniae* strain was assessed via MIC analysis. As shown in Figure 8, the results revealed that r*Lc*IL-8 exhibited MIC values ranging from 6.25 to 12.50 μg/mL, with an average value of 10.42 ± 3.61 µg/mL. Furthermore, when examining the bacterial growth rate in CFU/mL units, an inversely proportional relationship was observed between the increased protein concentration and the decreased number of bacteria. A decline in bacterial growth was evident at a protein concentration of 0.39 µg/mL, with 6.75 ± 0.15 Log CFU/mL (*p* < 0.05). However, the decrease in the bacterial growth rate tended to remain constant when exposed to protein concentrations ranging from 12.50 to 100.0 µg/mL.

### 3.8. Anti-S. iniae Effect of the rLcIL-8 Protein against S. iniae

The survival rates of fish treated with r*Lc*IL-8 protein for 14 days after challenge with *S. iniae* are shown in Figure 9. It is evident that all the concentrations of r*Lc*IL-8 resulted in increased survival rates after exposure to the virulent fish pathogen. Specifically, fish that received 50 and 100 µg of r*Lc*IL-8 showed strong resistance to infection, with significantly higher survival rates compared to the control group, which had survival rates of 93.3 ± 0.0% and 83.4 ± 1.7%, respectively (*p* < 0.001). In addition, fish injected with 10 µg of r*Lc*IL-8 during this period also exhibited significantly higher survival than the control fish at 70.0 ± 2.8% (*p* < 0.05). However, starting from day 5, the survival rate in each group stabilized and remained consistent until the end of the 14-day observation period.

## 4. Discussion

In this study, we successfully cloned and characterized cDNA encoding the mature interleukin-8 protein of Asian seabass (*LcIL-8*). The deduced *Lc*IL-8 sequence exhibited the characteristic arrangement of four cysteine residues (C30, C32, C56, and C73) commonly found in other vertebrate IL-8s. These four conserved cysteine residues played a crucial role in the construction of the tertiary protein structure and further influenced the crucial functions of the IL-8 molecules [29,30]. In addition, another typical feature of the *IL-8* gene is its arrangement of amino acids in the CXC pattern, also referred to as the CXC chemokine. In the case of *LcIL-8*, we observed that two of the cysteines at the N-terminal of CXC chemokine molecules were sequestered by a single non-conserved Arg31 and connected by disulfide bonds. Basically, CXC chemokine molecules can be structurally classified into two groups based on the tripeptide motif ELR appearance [31]. All mammalian IL-8 molecules are similarly identified to possess the ELR component; however, the ELR structure notably disappears in most teleosts, except for in Atlantic cod and haddock [26,32]. Our sequence analysis revealed an ILR motif upstream of the CXC motif in the deduced *Lc*IL-8 protein, where leucine replaced glutamic acid. Such an amino acid substitution has never been reported in studies of other fish species. Despite the replacement of the ELR motif, which is responsible for recruiting neutrophils and promoting angiogenesis in mammals [18,33], with ILR or other motifs, the recombinant proteins of some fish IL-8 variants continue to demonstrate chemotactic activity on neutrophils and macrophages [34,35]. Therefore, *LcIL-8* may represent an alternative form of ILR with similar functions. Nevertheless, the exact influence of each motif on the chemoattractant activity in the teleost IL-8 molecule still remains for the next investigation.

The amino acid sequence and phylogenetic analyses of the 37 IL-8 members obtained from fish and higher vertebrates revealed the presence of three distinct clades. Specifically, clades 2 and 3 exhibited associations with the IL-8 group found in teleost fish. Notably, within fish clade 2 of IL-8, a further distinction could be separated into two subgroups. It is known that teleosts have at least two conserved IL-8 lineages, in contrast to mammals, which possess only one IL-8 lineage, consistent with previous reports indicating the presence of two IL-8 lineages in fish [36,37]. These two IL-8 lineages are believed to have deviated by virtue of a chromosome duplication event in a teleost ancestor [38]. Among these, *Lc*IL-8 was clustered within the first subgroup of the second clade and exhibited close evolutionary relationships with yellowfin seabream and yellow croaker. Intriguingly, the IL-8 from mammals, birds, Western clawed frogs, cartilaginous fish, and European seabass IL-8s were all grouped within the same clade. This observation suggests the possibility that multiple evolutionary events may have occurred in both teleosts and higher vertebrates.

The tissue distribution of *LcIL-8* mRNA in healthy Asian seabass was analyzed via qRT–PCR. The results of the qRT–PCR demonstrated ubiquitous expression in all the tested tissues, which was consistent with reports in several fish, such as Atlantic cod [26], Japanese sea perch [39] and blunt snout bream [40]. Additionally, the highest *LcIL-8* expression level was observed in the gills, consistent with results from previous studies on *IL-8* in other teleost species [41,42], where *IL-8* genes were predominantly expressed in the gills, a common target tissue of many pathogens. This obtained information suggests that the IL-8 molecule may have a functional role in the host defense system by promoting the migration of local immune cells, including neutrophils, lymphocytes, monocytes, and macrophages, to inflammatory locations and their succeeding attachment to various endothelia [43,44]. Conversely, the liver, which exhibited relatively weak expression of *LcIL-8*, was found to have high expression levels in many other teleosts, such as half-smooth tongue sole [45], large yellow croaker [37,46] and South American fish [47]. These results underscore the highly inconsistent expression models found in different tissues between various fish species. As a result, it could be implied that *IL-8* cytokines possess complicated functional roles in homeosis in both nonimmune and immune tissues due to their species-specific patterns in teleost fish.

The route of infection is known to have a significant impact on the host–pathogen relationship. Consequently, a profound knowledge of the relationship between the immune defense mechanisms of fish and the different infection routes is necessary. In our current study, we analyzed the expression of *LcIL-8* in response to both internal infections by *S. iniae* and external infection by *F. covae* using qRT–PCR in fish immune-related organs. Most of these act as the major lymphoid cells producing organs in teleosts and serve as the primary root of local macrophages necessitated in bacterial phagocytic activity in the nonspecific immune system [48]. The results from this study showed that *LcIL-8* was significantly increased in all the tested tissues after exposure to both pathogenic bacteria, suggesting that these studied tissues can be the primary components of immune function cells to react to these pathogens. Notably, the highest expression was observed in whole blood after injection with 1 × 10^5^ CFU/fish *S. iniae*. This tissue contains numerous immune cells, including lymphocytes, monocytes, and phagocytes, which function as the major sources responsible for producing and releasing the cytokines that play a crucial role in the chemoattractant functions of various kinds of immune responsive cells [49]. With the *F. covae* challenge, a high level of *LcIL-8* mRNA was observed in the gills at 12 h in fish infected with 1 × 10^6^ CFU/mL *F. covae*. This obtained information demonstrates the fact that *S. iniae* can more systemically induce defense mechanisms than *F. covae*, which prefers to utterly occupy the exterior contents of the fish, such as the skin and gills, during the initially phase of infection. In addition, significant upregulation of *LcIL-8* mRNA was also observed in the intestine and liver reacted on *F. covae*. In teleosts, the liver is one of the most important hematopoietic tissues and is crucial for producing acute phase proteins (APPs), such as C-reactive protein (CRP), ceruloplasmin and transferrin, which are controlled various cytokines liberated in the inflammatory response [50]. Furthermore, epithelial cells in the intestine play a crucial role in producing IL-8 in response to invasive agents [51], and this may be one of the initial signals of acute mucosal inflammation in bacterial infections [52]. Therefore, the strong induction of *LcIL-8* after exposure to both pathogenic bacteria in this study indicates its potential proinflammatory role as a chemoattractant, recruiting inflammatory cells to the infected site.

Moreover, in the early stages, at 6–24 h following injection with *S. iniae* and 6–12 h following immersion with *F. covae*, *LcIL-8* mRNA displayed varying levels of upregulation in all the tested tissues, suggesting that during this critical period, *IL-8* mRNAs may be produced vigorously by various cells, including monocytes, macrophages and endothelial cells, in the infected site to eliminate foreign antigens [53]. These results align with those of previous studies in which fish IL-8s were observed to be regulated in response to various stimuli, including bacteria, LPS, and polyinosinic-polycytidylic acid (poly I:C). For example, large yellow croaker exhibited a response to LPS, poly I:C and *Vibrio parahaemolyticus* [37], Japanese flounder after stimulation with LPS [54] and Nile tilapia challenged with two pathogens of *S. agalactiae* and *Aeromonas hydrophila* [55].

However, the expression of the *LcIL-8* gene was downregulated and reached basal levels from 48 to 96 h across all the bacterial concentrations of *S. iniae*. This suggests that *IL-8* genes possess a brief/short half-life reacted on these two pathogens due to the AU-rich elements (AREs) located in the 3′ UTR, which play a critical role in terminating the response [56]. During *F. covae* challenge, *LcIL-8* mRNA expression in most tested tissues, except the gills, gradually decreased and was restored to the basal level from 24 to 96 h postinfection at all the bacterial concentrations. This suggests an abeyant function of *IL-8* cytokines in their transcriptional process by negative inhibition feedback via compatible protein receptors on the plasma membrane [57]. Thus, they exhibit rapid responses in the early stages of infection and subsequently degrade later. Furthermore, the regulatory and inhibitory mechanisms of other cytokines are essential since excessive expression of *IL-8* genes is directly associated with host tissue damage [58].

Interestingly, in the gills, the expression of *LcIL-8* levels in response to all the concentrations of *F. covae* was suppressed at 24 to 48 h, consistent with the mRNA levels of *LcIL-8* in the other mentioned tissues. However, at 96 h, this gene was upregulated once again, indicating that the gills are the primary target organs during a natural route infection. These gills not only function as physical barriers against environmental hazards but also serve as gill-associated lymphoid tissue (GIALT) to contain leucocyte populations and produce inflammatory cytokines, including tumor necrosis factor α (TNFα) and interleukins (ILs) [59,60]. It is plausible that the bacteria attempt to breach the gills to enter the bloodstream and initiate a systemic infection [61]. Thus, certain innate immune components within the gills may require longer periods to eliminate *F. covae* in this experiment.

Based on dose-dependent induction, it was observed that the highest concentration of both bacterial solutions strongly induced *LcIL-8* mRNA, followed by the moderate dose, while the lower concentration showed weaker induction. Similar results were observed in Nile tilapia after being induced with the highest dose of 1 × 10^9^ CFU/mL *S. agalactiae*, which potentially stimulated the expression of CXC chemokine correlated to lower bacterial concentrations [62]. These findings suggest that a higher bacterial concentration can enhance their ability to effectively spread, colonize, and invade host cells and tissues [63], which are together considered one of the primary factors leading to disease in the host.

In the current research, the recombinant *Lc*IL-8 protein was produced to assess its biological functions. The chemotactic activity of various r*Lc*IL-8 proteins has been widely studied in a variety of fish species, including black seabream [64], half-smooth tongue sole [45], Mandarin fish [65], and blunt snout bream [40]. Additionally, in this study, r*Lc*IL-8 was used to perform a phagocytic assay, given that phagocytosis is another important role of most chemokine proteins. Phagocytosis is fundamental for host defense against invading pathogens and plays a crucial role in immune and inflammatory responses [66]. The results of this study demonstrated a substantial increase in the phagocytic activity (PA) of Asian seabass phagocytes at all the concentrations (1, 10, and 100 µg) of r*Lc*IL-8. The highest PA level was observed in the treatment group that obtained 100 µg of r*Lc*IL-8, demonstrating the effectiveness of all the r*Lc*IL-8 concentrations in enhancing in vitro phagocytic activity. This obtained information is similar to the previous findings [62], demonstrating that phagocytes exposed to r*On*-CXC1 and r*On*-CXC2 at 1 and 10 µg/mL significantly enhance the PA of PBLs in Nile tilapia. Additionally, the treatment group that obtained 10 µg/mL of both r*On*-CXC1 and r*On*-CXC2 exhibited the highest level of PI. In the current investigation, the highest PI was achieved in the treatments exposed to 10 and 100 µg of r*Lc*IL-8. This suggests that both concentrations can stimulate phagocytes by enhancing their engulfment capability, while a concentration of 1 µg is more effective at enhancing the number of phagocytic cells rather than improving the antigen engulfment.

To determine the efficacy of *Lc*IL-8 in mediating antimicrobial activity against pathogenic *S. iniae*, an MIC assay was employed. The results revealed that 10.42 ± 3.61 µg/mL r*Lc*IL-8 protein is the minimum concentration needed to inhibit the growth of *S. iniae*. Furthermore, the inhibitory effect was observed in a dose-dependent manner. These results are similar to those found in other fish, where the derived peptide from the end of the C-terminal IL-8 protein was also determined via antibacterial assay, due to generating IL-8 by cutting at the N-terminus leading to more biologically active forms. Sáenz-Martínez et al. (2021) [67] reported that 10 μM *Om*IL-8α_80-97_ synthetic peptide from rainbow trout produced a 50% inhibition of bacterial growth in all the tested bacterial strains, including *Y. ruckeri*, *A. salmonicida*, *Pseudomonas aeruginosa*, *Staphylococcus aureus* and *E. coli*. Additionally, the synthetic peptide fragment WS12 of snakehead murrel (*Channa striata*) IL-8 at concentrations between 3.125 and 50 μM has been demonstrated antibacterial activity against *Bacillus cereus* and *E. coli* [68]. Previous studies on humans have shown that the IL-8-derived peptide exhibits antimicrobial activity against certain gram-positive and gram-negative bacteria [69]. These findings suggest that IL-8 may demonstrate an alternative biofunction as an antimicrobial molecule through direct action or the C-terminal end-derived peptide.

Our previous results indicated the additional significant ability of r*Lc*IL-8, which enhances resistance against virulent *S. iniae*. pathogen. The increased survival rates at all the protein concentrations after exposure to *S. iniae* indicate the protein’s efficacy in protecting Asian seabass against this harmful pathogen. This protective mechanism is supported by its capacity to inhibit *S. iniae* in vitro and improve antigen engulfment efficiency. Furthermore, another supporting result, as reported by Li et al. (2023) [55], in vitro investigations showed that r*On*-IL8 has complicate immune-effector functions in the head kidney lymphocytes, including upregulation of MyD88 and STAT3, downregulation of P38 and P65, and promotion of the apoptosis and inflammatory response. However, the study by Wang et al. (2016) [70] in channel catfish demonstrated that the cumulative survival of fish intramuscularly injected with *Pc*IL-8 for 4 weeks and then challenged by *S. iniae* resulted in a low survival rate of 20.0%, which did not statistically differ from the *Pc*IL-8 untreated group. This information implies that the protective effects of IL-8 proteins in fish against invading harmful bacteria may rely on (1) the concentration of IL-8, (2) the fish species, (3) the type, source and isoform of IL-8 molecule, and (4) the method and duration of protein administration to fish, which may involve different protective mechanisms.

## 5. Conclusions

In summary, *Lc*IL-8 was successfully cloned and identified for its structural molecule and profound evolution compared with IL-8s of various vertebrates. The obtained information from the qRT–PCR analysis information obtained in this study demonstrates the significant responses of *LcIL-8* to two severely pathogenic *S. iniae* and *F. covae*. Additionally, the r*Lc*IL-8 protein efficiently enhances the phagocytic activity of Asian seabass phagocytes. Importantly, r*Lc*IL-8 serves as a mediator of antimicrobial activity in vitro and perfectly exhibits its ability to protect fish from streptococcosis, which severely affects the Asian seabass aquaculture industry. Therefore, the insights obtained in this study are crucial for developing effective strategies, including highly potent vaccine adjuvants for preventing diseases caused by harmful pathogenic bacteria in Asian seabass aquaculture.

## Figures and Tables

**Figure 1 animals-14-00475-f001:**
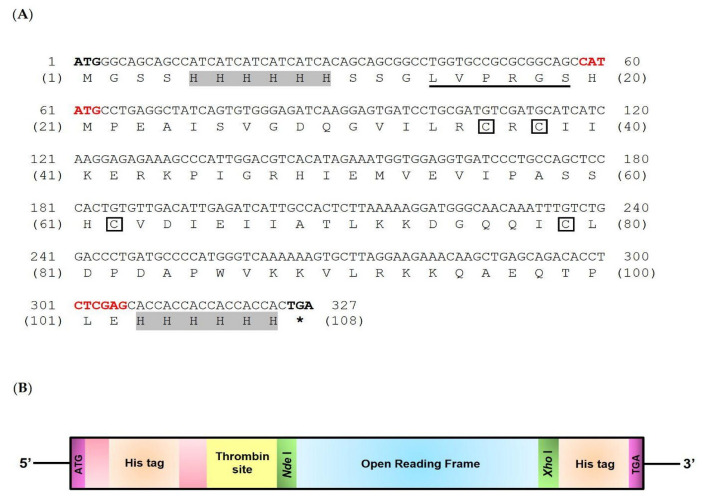
Structure of the mature cDNA encoding the *LcIL-8* gene used to produce the *Lc*IL-8 protein (**A**) and a schematic structure of the recombinant *Lc*IL-8 protein (r*Lc*IL-8) (**B**). The upper and lower numbers in (**A**) indicate the nucleotide and amino acid sequences, respectively. Gray shadows, black boxes and asterisks indicate 6-His tags, conserved cysteines and a stop codon, respectively. Red alphabets respectively indicate *Nde* I and *Xho* I double-restriction digestion sites.

**Figure 2 animals-14-00475-f002:**
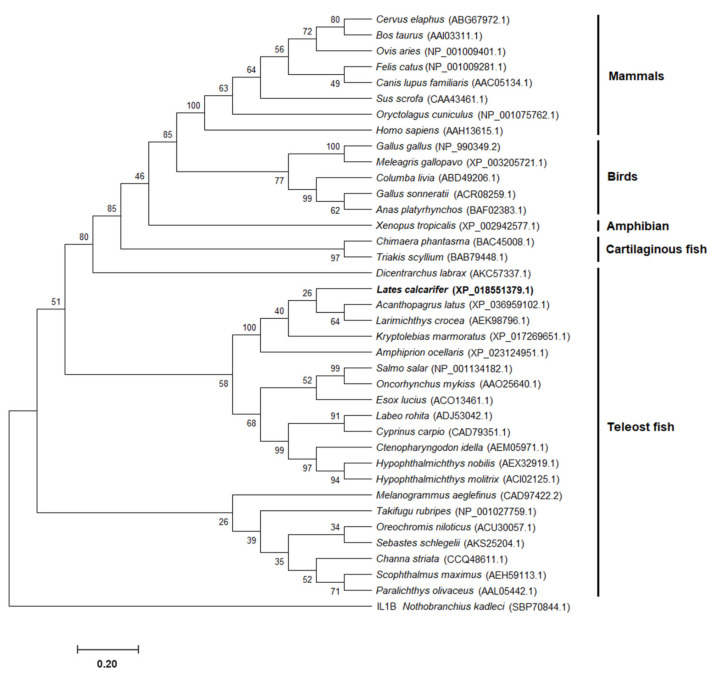
Phylogenetic analysis of *Lc*IL-8 and other IL-8 proteins of various vertebrates. The accession numbers of each IL-8 are indicated in parentheses.

**Figure 3 animals-14-00475-f003:**
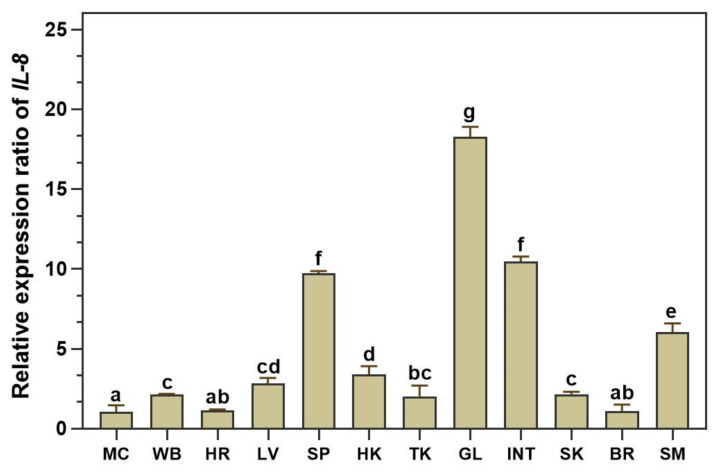
qRT–PCR assay of *LcIL-8* transcripts in various tissues. MC; muscle, WB; whole blood, HR; heart, LV; liver, SP; spleen, HK; head kidney, TK; trunk kidney; GL; gills; INT; intestine; SK; skin, BR; brain and SM; stomach. The different letters on each bar indicate statistically significant differences (*p* < 0.05).

**Figure 4 animals-14-00475-f004:**
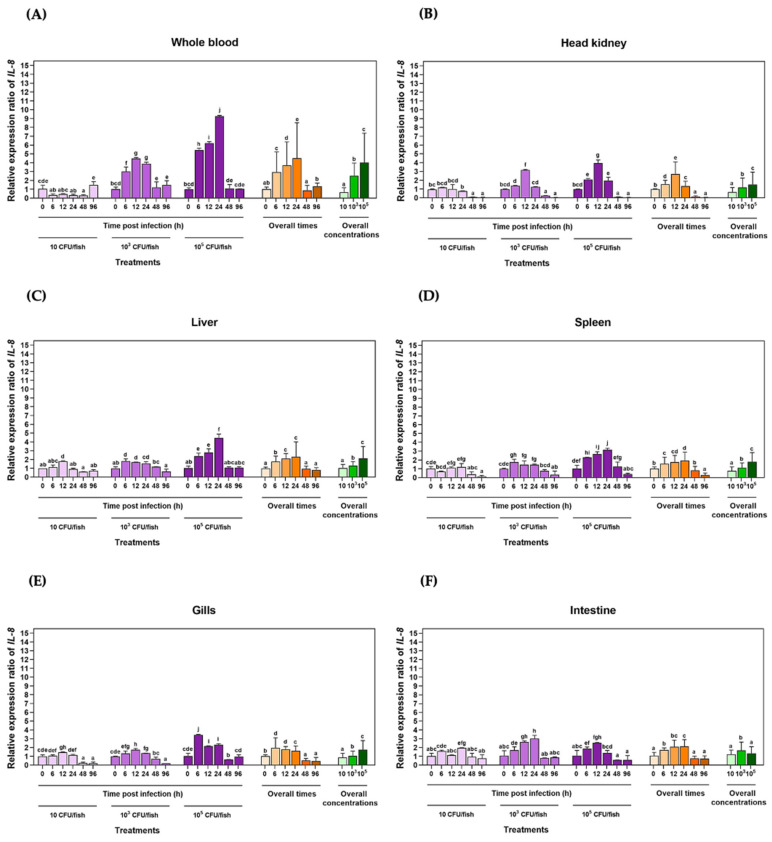
qRT–PCR analysis of *LcIL-8* transcripts in whole blood (**A**), head kidney (**B**), liver (**C**), spleen (**D**), gills (**E**) and intestine (**F**) of Asian seabass injected with different concentrations of *S. iniae* at different time points. The different letters on each bar indicate statistically significant differences (*p* < 0.05).

**Figure 5 animals-14-00475-f005:**
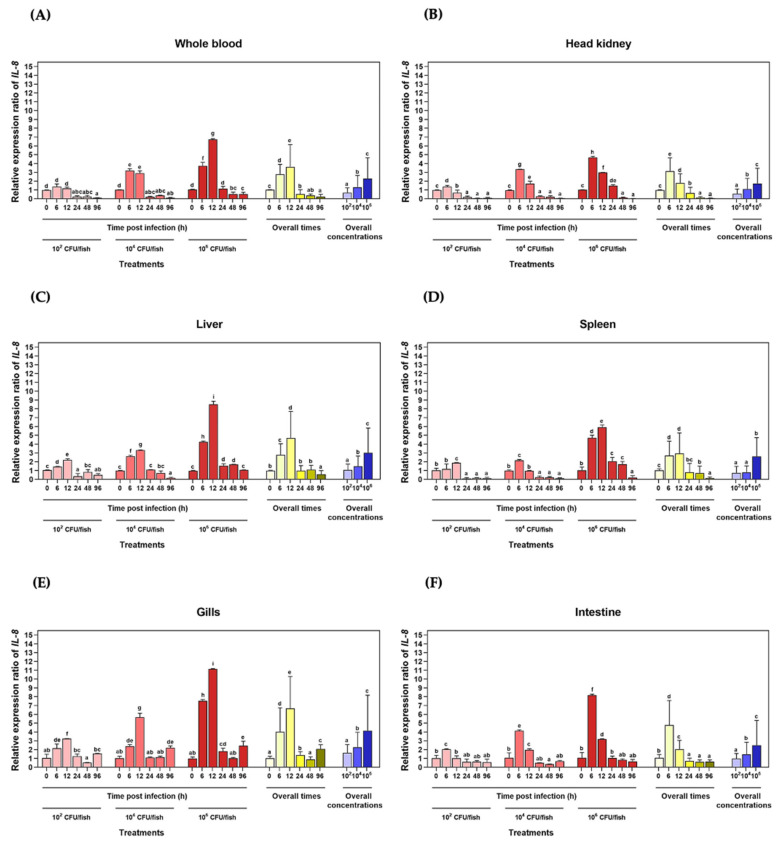
qRT–PCR assay of the *LcIL-8* mRNA levels in whole blood (**A**), head kidney (**B**), liver (**C**), spleen (**D**), gills (**E**) and intestine (**F**) of Asian seabass immersed with different concentrations of *F. covae* at different time points. The different letters on each bar indicate statistically significant differences (*p* < 0.05).

**Figure 6 animals-14-00475-f006:**
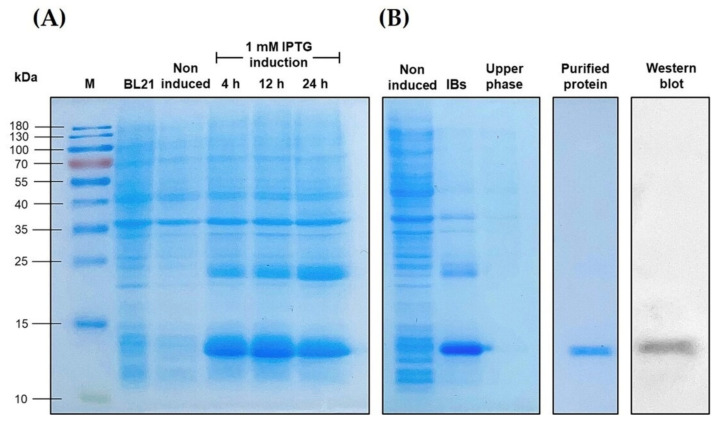
SDS–PAGE analysis of the r*LcI*L-8 protein. Induction analysis of r*Lc*IL-8 at different time points (**A**). M: protein marker; BL21: protein form BL21 cells; Noninduced: noninduced BL21 cells containing pET28b (+) r*Lc*IL-8; Lane 4–6: BL21 cells containing pET28b (+) r*Lc*IL-8 induced by 1 mM IPTG at 4, 12 and 24 h, respectively. Characterization of the inclusion bodies (IBs) r*Lc*IL-8 and its upper phase in the overexpression experiment and purification of the r*Lc*IL-8 and Western blot analysis of refolding r*Lc*IL-8 (**B**).

**Figure 7 animals-14-00475-f007:**
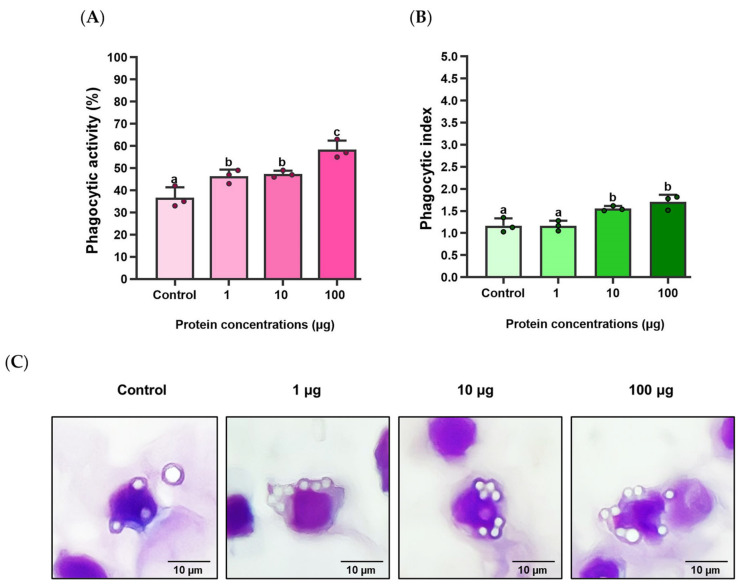
Percent phagocytic activity (**A**), phagocytic index (**B**) and phagocytic cell pictures (**C**) of Asian seabass phagocytes induced by various concentrations of r*Lc*IL-8. The different letters on each bar indicate statistically significant differences (*p* < 0.05).

**Figure 8 animals-14-00475-f008:**
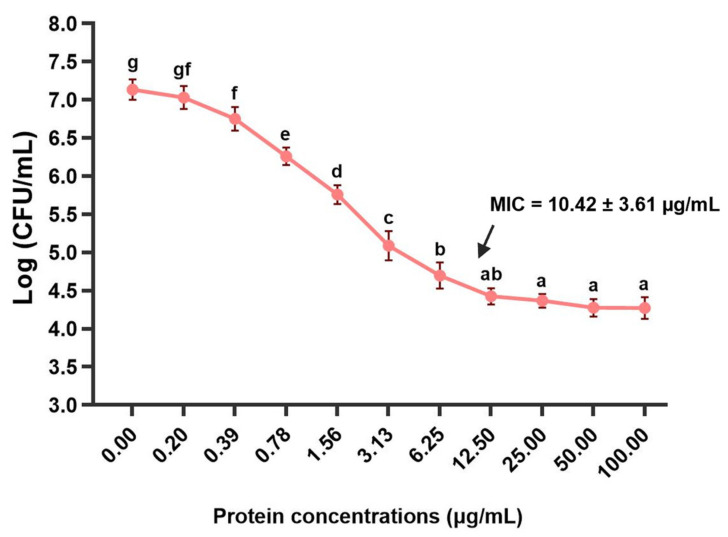
MIC analysis of *S. iniae* exposed to various concentrations of r*Lc*IL-8. The black arrow indicates the average MIC value. The different letters indicate significant differences (*p* < 0.05).

**Figure 9 animals-14-00475-f009:**
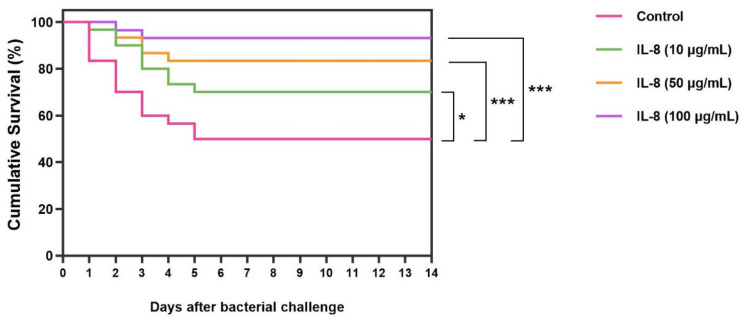
Survival analysis of Asian seabass first injected with *S. iniae* following secondary injection with various concentrations of r*Lc*IL-8. Asterisks “*” and “***”, respectively, demonstrate significant differences at *p* < 0.05 and *p* < 0.001.

**Table 1 animals-14-00475-t001:** Specific primers used for the PCR analysis in the current study.

Primer Name	Sequence (5′-3′)	Amplicon Size
*Lc*IL8 F	CATATGCCTGAGGCTATCAGTGTGGGAGAT	237 bp
*Lc*IL8 R	CTCGAGAGGTGTCTGCTCAGCTTGTTTCTT	
*Lc*IL-8 qF	TGATCCTGCGATGTCGATGCAT	206 bp
*Lc*IL-8 qR	AGGTGTCTGCTCAGCTTGTTTC	
*Lc-*β-actin qF	TACCCCATTGAGCACGGTATTG	150 bp
*Lc-*β-actin qF	TCTGGGTCATCTTCTCCCTGTT	

## Data Availability

The data that support the findings of this study are available on request from the corresponding authors.

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
