# Peer review of "Identification, Expression and Antimicrobial Functional Analysis of Interleukin-8 (IL-8) in Response to Streptococcus iniae and Flavobacterium covae in Asian Seabass (Lates calcarifer Bloch, 1790)"

_animals, 2024, doi:10.3390/ani14030475_

Round 1

Reviewer 1 Report

Comments and Suggestions for Authors

Line 26) The abstract is results oriented and should introduce the framework of the study.

Line 43) Is there any data regarding the volume of production ?

Line 83) The term "white blood cells" is a broad formulation and it would be more relevant to cite adequate immune cells (e.g. neutrophils; macrophages).

Line 104) The water temperature in fish fiberglass tank is missing.

Line 106) Is "healthy" considered as equal as "specific pathogen free" ? If yes, SPF would be clearer.

Line 186) The primer (F and R) concentration (µM) in the final reaction volume is missing.

Line 198) The PCR efficiency of the primer sets is not indicated. This information would be valuable in the Table 1.

Line 223) The sort form of "hours" with "hs" is not used after the first mention and in the rest of the manuscript. Therefore, I recommend to remove it.

Line 422) Did the normality of the datasets for ANOVA was assessed ? If yes, indicate the type of test (e.g. Shapiro-Wilk).

Line 570) The mention of Figure 4A is missing.

Line 576) Please mention the relevant Figure with the first mention of the results in the text (e.g. "in the head kidney (Figure 4B)").

Line 777) A brief reminder of the staining used for the phagocytes would be informative.

Line 782) Please mention Figure 8 here and not at the end of the results.

Line 808) Describe what the arrow is supposed to show. Also, missing mention "different letters indicate significant differences (p<0.05)".

Line 822) Please write IL-8 as such in the legend of the Figure 9 instead of IL8.

Author Response

Dear esteemed editor and reviewers

Firstly, we really appreciate all the comments suggested in our manuscripts. These comments are very useful for improving the quality of our scientific research to warrant an acceptable research document.

In the current version of our manuscript, we carefully correct and add necessary information by strictly following the valuable suggestions of the 3 anonymous reviewers, indicated by blue highlights as follows; 

Academic Editor Notes

English needs some improvement (some of the sentences are not easy to understand).

Response: Our manuscript has been carefully polished by a native speaker scientist, who is keen on this field.

The introduction lacks references and a description of chemokines (difference with cytokines?). Moreover, the authors should cite the negative role of IL-8 (when overexpressed or uncontrolled) and the pathogenesis of the two diseases (way of transmission and target organs).

Response: Thank you for this comment. We have properly added this information in the “Introduction.”

The M&M section should be completely reorganized: too many repetitions (the RNA extraction is always the same; they don't need to repeat it). Some interesting data are missing: the age of the fish, the in vivo authorization (national or international), the assessment of the RNA quantity and quality, and whether the tissues/organs pooled or investigated individually (it is not clear). The molarity of primers, how were they designed (on the same exon?). The specificity can be assessed only by sequencing, the melting temperature test is not sufficient.

Response: We have clarified these unclear points in the “M&M”.

The viability of the injected bacteria was checked? Usually, the bacteria are not used at the end of a long incubation (18-24 h), but they are "refreshed" by inoculating them in fresh media and used after a few hours (2-3). The number of animals is not clear (200 animals in 6 groups is 33/group; but the authors mentioned 30 animals/group!). There is not a real control group untreated/infected, but a T0 (that is not equal). The authors made a phagocytosis test on PBMC (not the most adequate cells), which should be better on neutrophils.

Response: Thank you for these comments. We have carefully clarified these target points.

- Based on our experience, the two pathogenic bacteria are always vigorous after the selected virulence strains are grown in culture media for 18-24 hr.

- The number of fish was corrected to “200”.

- In our experiment, Lymphoprep was technically used to isolate PBLs, which are mostly monocytes. In fish, neutrophils are very difficult to isolate via this technique and their population is very low compared with the other phagocytes.        

The main function of IL-8 is the recruitment of leukocytes; why the authors didn't perform a chemotaxis test as a functional test with the recombinant protein?

Response: Thank you for this comment. We understand well these specific comments. To indicate the chemotactic property of rLcIL-8 protein, we chose phagocytic activity based on our system support in our lab.   

After the challenge why the authors didn't check the clearance of the pathogen (CFU in homogenate tissues/organs)?

Response: It’s not so easy to check the remaining bacterial number in fish tissues, since it’s very high variation. By the way, we always employ the steak plate method to confirm bacterial appearance in all infected fish.

It is not clear to me how IL-8 can directly inhibit bacterial growth. What's the meaning of that experiment?

Response: We have added the dual functions of IL-8 molecules in “Discussion”. Based on recent information, IL-8 molecules have been found to possess dual functions of both chemotactic and antimicrobial properties.

Finally, the statistical analysis misses the normality test of the parameters.

Response: Thank you for these comments. We have carefully clarified this concern in section 2.10.

Reviewer 1#

Line 26) The abstract is results-oriented and should introduce the framework of the study.

Response: The brief framework of the study was added to the abstract section in accordance with the reviewer's suggestion.

Line 43) Is there any data regarding the volume of production?

Response: Production commenced with a 100 mL bacterial culture solution, as detailed in Section 2.6.3.

Response: The SPF (Specific Pathogen-Free) status has been redefined in Section 2.1 for clarity.

Line 83) The term "white blood cells" is a broad formulation and it would be more relevant to cite adequate immune cells (e.g. neutrophils; macrophages).

Response: It has been revised in accordance with the reviewer's suggestions.

Line 104) The water temperature in fish fiberglass tank is missing.

Response: Temperature data has been added in the 2.1 section

Line 106) Is "healthy" considered as equal as "specific pathogen free" ? If yes, SPF would be clearer.

Response: The SPF (Specific Pathogen-Free) status has been redefined in Section 2.1 for clarity.

Line 189) The primer (F and R) concentration (µM) in the final reaction volume is missing.

Response: The primers concentration of the primers has been added in the 2.4.2 section

Line 199) The PCR efficiency of the primer sets is not indicated. This information would be valuable in the Table 1.

Response: The efficiency of qPCR of the primers has been added in the 2.4.2 section

Line 225) The sort form of "hours" with "hs" is not used after the first mention and in the rest of the manuscript. Therefore, I recommend to remove it.

Response:  We have already replaced the abbreviated term 'hour' with 'h' throughout the manuscripts.

Line 566) The mention of Figure 4A is missing.

Response: Figure 4A was revised, mentioned, and cited in the text.

Line 572) Please mention the relevant Figure with the first mention of the results in the text (e.g. "in the head kidney (Figure 4B)").

Response: Figure 4B was revised, mentioned, and cited in the text.

Line 825) Describe what the arrow is supposed to show. Also, missing mention "different letters indicate significant differences (p<0.05)".

Response: The arrow's meaning in Figure 8 has been redefined in the captions.

Line 830) Please mention Figure 8 here and not at the end of the results.

Response: Figure 8 has been mentioned in the text.

Line 853) Please write IL-8 as such in the legend of Figure 9 instead of IL8.

Response: In Figure 9, 'IL8' has been rewritten as 'IL-8'.

Reviewer 2#

In this Manuscript (MS), the interleukin-8 (IL-8) gene of the Asian seabass (Lates calcarifer) was cloned and named LcIL-8, and its expression levels were examined in normal and diseased fish tissues. The recombinant LcIL-8 protein (rLcIL-8) was produced and evaluated for its biological functions under different conditions.

 The manuscript presents intriguing findings, and I recommend its publication. However, there are some points that need to be addressed:

Abstract: This section should be enhanced to better elucidate the rationale of the study. It might be helpful to clearly distinguish recombinant Lc-IL8 from the natural form. Additionally, mention that the therapeutic effect was evaluated through recombinant protein injection.

Response: The abstract has been revised according to the suggestions provided in the manuscripts.

M&M: Some sections could be consolidated, especially those pertaining to the basic QPCR protocols. This consolidation would aid in focusing on the most critical aspects.

Response: The details of this part have been revised and redescribed in the manuscripts.

Results:

Figure 3. The y axis should be limited to 25.

Response: In Figure 3, the y-axis values have been adjusted to a maximum limit of 25.

Fig 4 & 5: it will be better with larger letter sizes.

Response: The size of the figure has been adjusted to fit within the manuscript

Discussion: It is OK.

Response: Thank you for your suggestions.

Reviewer 3#

The manuscript entitled (Identification, expression and antimicrobial functional analysis of interleukin-8 (IL-8) in response to Streptococcus iniae and Flavobacterium covae in Asian seabass (Lates calcarifer Bloch, 5 1790)) is interesting and need a minor changes

1-the abstract should modified, the abstract should contain a sentence as an introduction to the subject of the study, and then briefly mentioned the methodology, then the main findings.

Response: The abstract has been revised according to the suggestions provided in the manuscripts.

2-the introduction is interesting and properly written

Response: Thank you for your valuable comments.

3-section 2.1., the water parameters should be measured and mentioned such as dissolved oxygen, temp, nitrite, ammonia, and light: dark period

Response: The water quality data of this study was added in the section of 2.1.

4- Line 106, How the authors tested the fish health

Response: The virus was examined using qPCR to ensure the health of the fish before conducting the experiments.

5-Line 113, the authors should mention the protein level and energy of the diet at least

Response: The crude ingredients of the feed used in the experiment were added in Section 2.13

6-section 2.5.1., from which host the bacterial isolates (S. iniae and F. covae) were isolated host?

Response: The both bacterial isolates, S. iniae and F. covae, were originally isolated from the infection Asian seabass in Thailand during the disease outbreak.

And how the authors identified the isolates?. Did the isolates have an accession no on the gene bank?

Response: Yes, the isolations of both pathogens were identified by 16S rRNA gene sequencing, and the reference for the data was published by Tumree et al., 2023.

7-Line 226 the authors said (resulting in a concentration of 1×109  colony-forming unit (CFU)/mL), how the authors validate the results (I mean how they obtain this concentration).

Response: The concentration of each isolated bacterium was measured using a spectrophotometer at OD 600 nm and compared to the results from plate counting data of each isolate.

8- The weight of the samples used for RNA extraction should be mentioned

Response: The weight of the RNA used in extraction has been mentioned in Section 2.2 and 2.4.1

9- The resolutions of figure 4 and 5 is very poor and should be improved

Response: The new figure has been added to the manuscript.

10- the discussion and conclusion are properly written

Response: Thank you for your valuable comments.

Reviewer 2 Report

Comments and Suggestions for Authors

In this Manuscript (MS), the interleukin-8 (IL-8) gene of the Asian seabass (Lates calcarifer) was cloned and named LcIL-8, and its expression levels were examined in normal and diseased fish tissues. The recombinant LcIL-8 protein (rLcIL-8) was produced and evaluated for its biological functions under different conditions.

The manuscript presents intriguing findings, and I recommend its publication. However, there are some points that need to be addressed:

Abstract: This section should be enhanced to better elucidate the rationale of the study. It might be helpful to clearly distinguish recombinant Lc-IL8 from the natural form. Additionally, mention that the therapeutic effect was evaluated through recombinant protein injection.

M&M: Some sections could be consolidated, especially those pertaining to the basic QPCR protocols. This consolidation would aid in focusing on the most critical aspects.

Results:

Figure 3. The y axis should be limited to 25.

Fig 4 & 5: it will be better with larger letter sizes.

Discussion : It is OK.

Author Response

(The authors gave the same response as above.)

Academic Editor Notes

English needs some improvement (some of the sentences are not easy to understand).

Response: Our manuscript has been carefully polished by a native speaker scientist, who is keen on this field.

The introduction lacks references and a description of chemokines (difference with cytokines?). Moreover, the authors should cite the negative role of IL-8 (when overexpressed or uncontrolled) and the pathogenesis of the two diseases (way of transmission and target organs).

Response: Thank you for this comment. We have properly added this information in the “Introduction.”

The M&M section should be completely reorganized: too many repetitions (the RNA extraction is always the same; they don't need to repeat it). Some interesting data are missing: the age of the fish, the in vivo authorization (national or international), the assessment of the RNA quantity and quality, and whether the tissues/organs pooled or investigated individually (it is not clear). The molarity of primers, how were they designed (on the same exon?). The specificity can be assessed only by sequencing, the melting temperature test is not sufficient.

Response: We have clarified these unclear points in the “M&M”.

The viability of the injected bacteria was checked? Usually, the bacteria are not used at the end of a long incubation (18-24 h), but they are "refreshed" by inoculating them in fresh media and used after a few hours (2-3). The number of animals is not clear (200 animals in 6 groups is 33/group; but the authors mentioned 30 animals/group!). There is not a real control group untreated/infected, but a T0 (that is not equal). The authors made a phagocytosis test on PBMC (not the most adequate cells), which should be better on neutrophils.

Response: Thank you for these comments. We have carefully clarified these target points.

- Based on our experience, the two pathogenic bacteria are always vigorous after the selected virulence strains are grown in culture media for 18-24 hr.

- The number of fish was corrected to “200”.

- In our experiment, Lymphoprep was technically used to isolate PBLs, which are mostly monocytes. In fish, neutrophils are very difficult to isolate via this technique and their population is very low compared with the other phagocytes.        

The main function of IL-8 is the recruitment of leukocytes; why the authors didn't perform a chemotaxis test as a functional test with the recombinant protein?

Response: Thank you for this comment. We understand well these specific comments. To indicate the chemotactic property of rLcIL-8 protein, we chose phagocytic activity based on our system support in our lab.   

After the challenge why the authors didn't check the clearance of the pathogen (CFU in homogenate tissues/organs)?

Response: It’s not so easy to check the remaining bacterial number in fish tissues, since it’s very high variation. By the way, we always employ the steak plate method to confirm bacterial appearance in all infected fish.

It is not clear to me how IL-8 can directly inhibit bacterial growth. What's the meaning of that experiment?

Response: We have added the dual functions of IL-8 molecules in “Discussion”. Based on recent information, IL-8 molecules have been found to possess dual functions of both chemotactic and antimicrobial properties.

Finally, the statistical analysis misses the normality test of the parameters.

Response: Thank you for these comments. We have carefully clarified this concern in section 2.10.

Reviewer 1#

Line 26) The abstract is results-oriented and should introduce the framework of the study.

Response: The brief framework of the study was added to the abstract section in accordance with the reviewer's suggestion.

Line 43) Is there any data regarding the volume of production?

Response: Production commenced with a 100 mL bacterial culture solution, as detailed in Section 2.6.3.

Response: The SPF (Specific Pathogen-Free) status has been redefined in Section 2.1 for clarity.

Line 83) The term "white blood cells" is a broad formulation and it would be more relevant to cite adequate immune cells (e.g. neutrophils; macrophages).

Response: It has been revised in accordance with the reviewer's suggestions.

Line 104) The water temperature in fish fiberglass tank is missing.

Response: Temperature data has been added in the 2.1 section

Line 106) Is "healthy" considered as equal as "specific pathogen free" ? If yes, SPF would be clearer.

Response: The SPF (Specific Pathogen-Free) status has been redefined in Section 2.1 for clarity.

Line 189) The primer (F and R) concentration (µM) in the final reaction volume is missing.

Response: The primers concentration of the primers has been added in the 2.4.2 section

Line 199) The PCR efficiency of the primer sets is not indicated. This information would be valuable in the Table 1.

Response: The efficiency of qPCR of the primers has been added in the 2.4.2 section

Line 225) The sort form of "hours" with "hs" is not used after the first mention and in the rest of the manuscript. Therefore, I recommend to remove it.

Response:  We have already replaced the abbreviated term 'hour' with 'h' throughout the manuscripts.

Line 566) The mention of Figure 4A is missing.

Response: Figure 4A was revised, mentioned, and cited in the text.

Line 572) Please mention the relevant Figure with the first mention of the results in the text (e.g. "in the head kidney (Figure 4B)").

Response: Figure 4B was revised, mentioned, and cited in the text.

Line 825) Describe what the arrow is supposed to show. Also, missing mention "different letters indicate significant differences (p<0.05)".

Response: The arrow's meaning in Figure 8 has been redefined in the captions.

Line 830) Please mention Figure 8 here and not at the end of the results.

Response: Figure 8 has been mentioned in the text.

Line 853) Please write IL-8 as such in the legend of Figure 9 instead of IL8.

Response: In Figure 9, 'IL8' has been rewritten as 'IL-8'.

Reviewer 2#

In this Manuscript (MS), the interleukin-8 (IL-8) gene of the Asian seabass (Lates calcarifer) was cloned and named LcIL-8, and its expression levels were examined in normal and diseased fish tissues. The recombinant LcIL-8 protein (rLcIL-8) was produced and evaluated for its biological functions under different conditions.

 The manuscript presents intriguing findings, and I recommend its publication. However, there are some points that need to be addressed:

Abstract: This section should be enhanced to better elucidate the rationale of the study. It might be helpful to clearly distinguish recombinant Lc-IL8 from the natural form. Additionally, mention that the therapeutic effect was evaluated through recombinant protein injection.

Response: The abstract has been revised according to the suggestions provided in the manuscripts.

M&M: Some sections could be consolidated, especially those pertaining to the basic QPCR protocols. This consolidation would aid in focusing on the most critical aspects.

Response: The details of this part have been revised and redescribed in the manuscripts.

Results:

Figure 3. The y axis should be limited to 25.

Response: In Figure 3, the y-axis values have been adjusted to a maximum limit of 25.

Fig 4 & 5: it will be better with larger letter sizes.

Response: The size of the figure has been adjusted to fit within the manuscript

Discussion: It is OK.

Response: Thank you for your suggestions.

Reviewer 3#

The manuscript entitled (Identification, expression and antimicrobial functional analysis of interleukin-8 (IL-8) in response to Streptococcus iniae and Flavobacterium covae in Asian seabass (Lates calcarifer Bloch, 5 1790)) is interesting and need a minor changes

1-the abstract should modified, the abstract should contain a sentence as an introduction to the subject of the study, and then briefly mentioned the methodology, then the main findings.

Response: The abstract has been revised according to the suggestions provided in the manuscripts.

2-the introduction is interesting and properly written

Response: Thank you for your valuable comments.

3-section 2.1., the water parameters should be measured and mentioned such as dissolved oxygen, temp, nitrite, ammonia, and light: dark period

Response: The water quality data of this study was added in the section of 2.1.

4- Line 106, How the authors tested the fish health

Response: The virus was examined using qPCR to ensure the health of the fish before conducting the experiments.

5-Line 113, the authors should mention the protein level and energy of the diet at least

Response: The crude ingredients of the feed used in the experiment were added in Section 2.13

6-section 2.5.1., from which host the bacterial isolates (S. iniae and F. covae) were isolated host?

Response: The both bacterial isolates, S. iniae and F. covae, were originally isolated from the infection Asian seabass in Thailand during the disease outbreak.

And how the authors identified the isolates?. Did the isolates have an accession no on the gene bank?

Response: Yes, the isolations of both pathogens were identified by 16S rRNA gene sequencing, and the reference for the data was published by Tumree et al., 2023.

7-Line 226 the authors said (resulting in a concentration of 1×109  colony-forming unit (CFU)/mL), how the authors validate the results (I mean how they obtain this concentration).

Response: The concentration of each isolated bacterium was measured using a spectrophotometer at OD 600 nm and compared to the results from plate counting data of each isolate.

8- The weight of the samples used for RNA extraction should be mentioned

Response: The weight of the RNA used in extraction has been mentioned in Section 2.2 and 2.4.1

9- The resolutions of figure 4 and 5 is very poor and should be improved

Response: The new figure has been added to the manuscript.

10- the discussion and conclusion are properly written

Response: Thank you for your valuable comments.

Reviewer 3 Report

Comments and Suggestions for Authors

The manuscript entitled (Identification, expression and antimicrobial functional analysis of interleukin-8 (IL-8) in response to Streptococcus iniae and Flavobacterium covae in Asian seabass (Lates calcarifer Bloch, 5 1790)) is interesting and need a minor changes

1-the abstract should modified, the abstract should contain a sentence as an introduction to the subject of the study, and then briefly mentioned the methodology, then the main findings.

2-the introduction is interesting and properly written

3-section 2.1., the water parameters should be measured and mentioned such as dissolved oxygen, temp, nitrite, ammonia, and light: dark period

4- Line 106, How the authors tested the fish health

5-Line 113, the authors should mention the protein level and energy of the diet at least

6-section 2.5.1., from which host the bacterial isolates (S. iniae and F. covae) were isolated host?

And how the authors identified the isolates?. Did the isolates have an accession no on the gene bank?

7-Line 226 the authors said (resulting in a concentration of 1×109  colony-forming unit (CFU)/mL), how the authors validate the results (I mean how they obtain this concentration).

8- The weight of the samples used for RNA extraction should be mentioned

9- The resolutions of figure 4 and 5 is very poor should improved

10- the discussion and conclusion are properly written

Author Response

(The authors gave the same response as above.)

Academic Editor Notes

English needs some improvement (some of the sentences are not easy to understand).

Response: Our manuscript has been carefully polished by a native speaker scientist, who is keen on this field.

The introduction lacks references and a description of chemokines (difference with cytokines?). Moreover, the authors should cite the negative role of IL-8 (when overexpressed or uncontrolled) and the pathogenesis of the two diseases (way of transmission and target organs).

Response: Thank you for this comment. We have properly added this information in the “Introduction.”

The M&M section should be completely reorganized: too many repetitions (the RNA extraction is always the same; they don't need to repeat it). Some interesting data are missing: the age of the fish, the in vivo authorization (national or international), the assessment of the RNA quantity and quality, and whether the tissues/organs pooled or investigated individually (it is not clear). The molarity of primers, how were they designed (on the same exon?). The specificity can be assessed only by sequencing, the melting temperature test is not sufficient.

Response: We have clarified these unclear points in the “M&M”.

The viability of the injected bacteria was checked? Usually, the bacteria are not used at the end of a long incubation (18-24 h), but they are "refreshed" by inoculating them in fresh media and used after a few hours (2-3). The number of animals is not clear (200 animals in 6 groups is 33/group; but the authors mentioned 30 animals/group!). There is not a real control group untreated/infected, but a T0 (that is not equal). The authors made a phagocytosis test on PBMC (not the most adequate cells), which should be better on neutrophils.

Response: Thank you for these comments. We have carefully clarified these target points.

- Based on our experience, the two pathogenic bacteria are always vigorous after the selected virulence strains are grown in culture media for 18-24 hr.

- The number of fish was corrected to “200”.

- In our experiment, Lymphoprep was technically used to isolate PBLs, which are mostly monocytes. In fish, neutrophils are very difficult to isolate via this technique and their population is very low compared with the other phagocytes.        

The main function of IL-8 is the recruitment of leukocytes; why the authors didn't perform a chemotaxis test as a functional test with the recombinant protein?

Response: Thank you for this comment. We understand well these specific comments. To indicate the chemotactic property of rLcIL-8 protein, we chose phagocytic activity based on our system support in our lab.   

After the challenge why the authors didn't check the clearance of the pathogen (CFU in homogenate tissues/organs)?

Response: It’s not so easy to check the remaining bacterial number in fish tissues, since it’s very high variation. By the way, we always employ the steak plate method to confirm bacterial appearance in all infected fish.

It is not clear to me how IL-8 can directly inhibit bacterial growth. What's the meaning of that experiment?

Response: We have added the dual functions of IL-8 molecules in “Discussion”. Based on recent information, IL-8 molecules have been found to possess dual functions of both chemotactic and antimicrobial properties.

Finally, the statistical analysis misses the normality test of the parameters.

Response: Thank you for these comments. We have carefully clarified this concern in section 2.10.

Reviewer 1#

Line 26) The abstract is results-oriented and should introduce the framework of the study.

Response: The brief framework of the study was added to the abstract section in accordance with the reviewer's suggestion.

Line 43) Is there any data regarding the volume of production?

Response: Production commenced with a 100 mL bacterial culture solution, as detailed in Section 2.6.3.

Response: The SPF (Specific Pathogen-Free) status has been redefined in Section 2.1 for clarity.

Line 83) The term "white blood cells" is a broad formulation and it would be more relevant to cite adequate immune cells (e.g. neutrophils; macrophages).

Response: It has been revised in accordance with the reviewer's suggestions.

Line 104) The water temperature in fish fiberglass tank is missing.

Response: Temperature data has been added in the 2.1 section

Line 106) Is "healthy" considered as equal as "specific pathogen free" ? If yes, SPF would be clearer.

Response: The SPF (Specific Pathogen-Free) status has been redefined in Section 2.1 for clarity.

Line 189) The primer (F and R) concentration (µM) in the final reaction volume is missing.

Response: The primers concentration of the primers has been added in the 2.4.2 section

Line 199) The PCR efficiency of the primer sets is not indicated. This information would be valuable in the Table 1.

Response: The efficiency of qPCR of the primers has been added in the 2.4.2 section

Line 225) The sort form of "hours" with "hs" is not used after the first mention and in the rest of the manuscript. Therefore, I recommend to remove it.

Response:  We have already replaced the abbreviated term 'hour' with 'h' throughout the manuscripts.

Line 566) The mention of Figure 4A is missing.

Response: Figure 4A was revised, mentioned, and cited in the text.

Line 572) Please mention the relevant Figure with the first mention of the results in the text (e.g. "in the head kidney (Figure 4B)").

Response: Figure 4B was revised, mentioned, and cited in the text.

Line 825) Describe what the arrow is supposed to show. Also, missing mention "different letters indicate significant differences (p<0.05)".

Response: The arrow's meaning in Figure 8 has been redefined in the captions.

Line 830) Please mention Figure 8 here and not at the end of the results.

Response: Figure 8 has been mentioned in the text.

Line 853) Please write IL-8 as such in the legend of Figure 9 instead of IL8.

Response: In Figure 9, 'IL8' has been rewritten as 'IL-8'.

Reviewer 2#

In this Manuscript (MS), the interleukin-8 (IL-8) gene of the Asian seabass (Lates calcarifer) was cloned and named LcIL-8, and its expression levels were examined in normal and diseased fish tissues. The recombinant LcIL-8 protein (rLcIL-8) was produced and evaluated for its biological functions under different conditions.

 The manuscript presents intriguing findings, and I recommend its publication. However, there are some points that need to be addressed:

Abstract: This section should be enhanced to better elucidate the rationale of the study. It might be helpful to clearly distinguish recombinant Lc-IL8 from the natural form. Additionally, mention that the therapeutic effect was evaluated through recombinant protein injection.

Response: The abstract has been revised according to the suggestions provided in the manuscripts.

M&M: Some sections could be consolidated, especially those pertaining to the basic QPCR protocols. This consolidation would aid in focusing on the most critical aspects.

Response: The details of this part have been revised and redescribed in the manuscripts.

Results:

Figure 3. The y axis should be limited to 25.

Response: In Figure 3, the y-axis values have been adjusted to a maximum limit of 25.

Fig 4 & 5: it will be better with larger letter sizes.

Response: The size of the figure has been adjusted to fit within the manuscript

Discussion: It is OK.

Response: Thank you for your suggestions.

Reviewer 3#

The manuscript entitled (Identification, expression and antimicrobial functional analysis of interleukin-8 (IL-8) in response to Streptococcus iniae and Flavobacterium covae in Asian seabass (Lates calcarifer Bloch, 5 1790)) is interesting and need a minor changes

1-the abstract should modified, the abstract should contain a sentence as an introduction to the subject of the study, and then briefly mentioned the methodology, then the main findings.

Response: The abstract has been revised according to the suggestions provided in the manuscripts.

2-the introduction is interesting and properly written

Response: Thank you for your valuable comments.

3-section 2.1., the water parameters should be measured and mentioned such as dissolved oxygen, temp, nitrite, ammonia, and light: dark period

Response: The water quality data of this study was added in the section of 2.1.

4- Line 106, How the authors tested the fish health

Response: The virus was examined using qPCR to ensure the health of the fish before conducting the experiments.

5-Line 113, the authors should mention the protein level and energy of the diet at least

Response: The crude ingredients of the feed used in the experiment were added in Section 2.13

6-section 2.5.1., from which host the bacterial isolates (S. iniae and F. covae) were isolated host?

Response: The both bacterial isolates, S. iniae and F. covae, were originally isolated from the infection Asian seabass in Thailand during the disease outbreak.

And how the authors identified the isolates?. Did the isolates have an accession no on the gene bank?

Response: Yes, the isolations of both pathogens were identified by 16S rRNA gene sequencing, and the reference for the data was published by Tumree et al., 2023.

7-Line 226 the authors said (resulting in a concentration of 1×109  colony-forming unit (CFU)/mL), how the authors validate the results (I mean how they obtain this concentration).

Response: The concentration of each isolated bacterium was measured using a spectrophotometer at OD 600 nm and compared to the results from plate counting data of each isolate.

8- The weight of the samples used for RNA extraction should be mentioned

Response: The weight of the RNA used in extraction has been mentioned in Section 2.2 and 2.4.1

9- The resolutions of figure 4 and 5 is very poor and should be improved

Response: The new figure has been added to the manuscript.

10- the discussion and conclusion are properly written

Response: Thank you for your valuable comments.